# Differentiation of Blackbox Combinatorial Solvers

**Marin Vlastelica**[1][*], **Anselm Paulus**[1][*], **Vít Musil**[2], **Georg Martius**[1], **Michal Rolínek**[1]

[1] Max-Planck-Institute for Intelligent Systems, Tübingen, Germany
[2] Università degli Studi di Firenze, Italy

{*marin.vlastelica, anselm.paulus, georg.martius, michal.rolinek*}*@tuebingen.mpg.de*
*vit.musil@unifi.it*

## Abstract

Achieving fusion of deep learning with combinatorial algorithms promises transformative changes to artificial intelligence. One possible approach is to introduce combinatorial building blocks into neural networks. Such end-to-end architectures have the potential to tackle combinatorial problems on raw input data such as ensuring global consistency in multi-object tracking or route planning on maps in robotics. In this work, we present a method that implements an efficient backward pass through blackbox implementations of combinatorial solvers with linear objective functions. We provide both theoretical and experimental backing. In particular, we incorporate the Gurobi MIP solver, Blossom V algorithm, and Dijkstra's algorithm into architectures that extract suitable features from raw inputs for the traveling salesman problem, the min-cost perfect matching problem and the shortest path problem. The code is available at

https://github.com/martius-lab/blackbox-backprop.

## 1 Introduction

The toolbox of popular methods in computer science currently sees a split into two major components. On the one hand, there are classical algorithmic techniques from discrete optimization – graph algorithms, SAT-solvers, integer programming solvers – often with heavily optimized implementations and theoretical guarantees on runtime and performance. On the other hand, there is the realm of deep learning allowing data-driven feature extraction as well as the flexible design of end-to-end architectures. The fusion of deep learning with combinatorial optimization is desirable both for foundational reasons – extending the reach of deep learning to data with large combinatorial complexity – and in practical applications. These often occur for example in computer vision problems that require solving a combinatorial sub-task on top of features extracted from raw input such as establishing global consistency in multi-object tracking from a sequence of frames.

The fundamental problem with constructing hybrid architectures is differentiability of the combinatorial components. State-of-the-art approaches pursue the following paradigm: introduce suitable approximations or modifications of the objective function or of a baseline algorithm that eventually yield a differentiable computation. The resulting algorithms are often sub-optimal in terms of runtime, performance and optimality guarantees when compared to their *unmodified* counterparts. While the sources of sub-optimality vary from example to example, there is a common theme: any differentiable algorithm in particular outputs continuous values and as such it solves a *relaxation* of the original problem. It is well-known in combinatorial optimization theory that even strong and practical convex relaxations induce lower bounds on the approximation ratio for large classes of problems (Raghavendra, 2008; Thapper & Živný, 2017) which makes them inherently sub-optimal. This inability to incorporate the best implementations of the best algorithms is unsatisfactory.

In this paper, we propose a method that, at the cost of one hyperparameter, implements a backward pass for a **blackbox implementation** of a combinatorial algorithm or a solver that optimizes a linear

---

[*]These authors contributed equally.

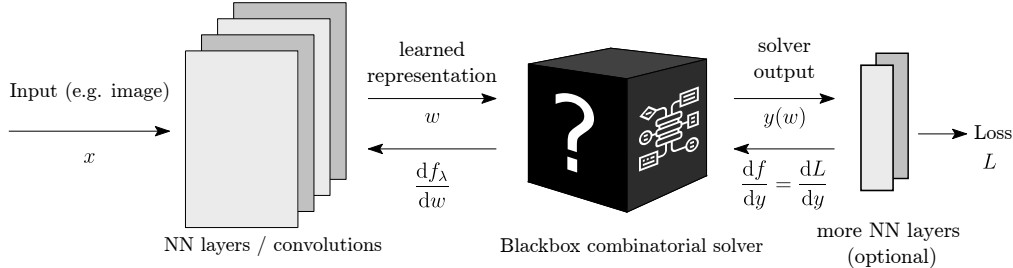

Figure 1: Architecture design enabled by Theorem 1. Blackbox combinatorial solver embedded into a neural network.

objective function. This effectively turns the algorithm or solver into a composable building block of neural network architectures, as illustrated in Fig. 1. Suitable problems with linear objective include classical problems such as SHORTEST-PATH, TRAVELING-SALESMAN (TSP), MIN-COST-PERFECT-MATCHING, various cut problems as well as entire frameworks such as integer programs (IP), Markov random fields (MRF) and conditional random fields (CRF).

The main technical challenge boils down to providing an informative gradient of a piecewise constant function. To that end, we are able to heavily leverage the minimization structure of the underlying combinatorial problem and efficiently compute a gradient of a continuous interpolation. While the roots of the method lie in loss-augmented inference, the employed mathematical technique for continuous interpolation is novel. The computational cost of the introduced **backward pass matches the cost of the forward pass**. In particular, it also amounts to one call to the solver.

In experiments, we train architectures that contain **unmodified implementations** of the following efficient combinatorial algorithms: general-purpose mixed-integer programming solver Gurobi (Gurobi Optimization, 2019), state-of-the-art C implementation of MIN-COST-PERFECT-MATCHING algorithm – Blossom V (Kolmogorov, 2009) and Dijkstra's algorithm (Dijkstra, 1959) for SHORTEST-PATH. We demonstrate that the resulting architectures train without sophisticated tweaks and are able to solve tasks that are beyond the capabilities of conventional neural networks.

## 2 RELATED WORK

Multiple lines of work lie at the intersection of combinatorial algorithms and deep learning. We primarily distinguish them by their motivation.

**Motivated by applied problems.** Even though computer vision has seen a substantial shift from combinatorial methods to deep learning, some problems still have a strong combinatorial aspect and require hybrid approaches. Examples include multi-object tracking (Schulter et al., 2017), semantic segmentation (Chen et al., 2018), multi-person pose estimation (Pishchulin et al., 2016; Song et al., 2018), stereo matching (Knöbelreiter et al., 2017) and person re-identification (Ye et al., 2017). The combinatorial algorithms in question are typically Markov random fields (MRF) (Chen et al., 2015), conditional random fields (CRF) (Marin et al., 2019), graph matching (Ye et al., 2017) or integer programming (Schulter et al., 2017). In recent years, a plethora of hybrid end-to-end architectures have been proposed. The techniques used for constructing the backward pass range from employing various relaxations and approximations of the combinatorial problem (Chen et al., 2015; Zheng et al., 2015) over differentiating a fixed number of iterations of an iterative solver (Paschalidou et al., 2018; Tompson et al., 2014; Liu et al., 2015) all the way to relying on the structured SVM framework (Tsochantaridis et al., 2005; Chen et al., 2015).

**Motivated by "bridging the gap".** Building links between combinatorics and deep learning can also be viewed as a foundational problem; for example, (Battaglia et al., 2018) advocate that "combinatorial generalization must be a top priority for AI". One such line of work focuses on designing architectures with algorithmic structural prior – for example by mimicking the layout of a Turing machine (Sukhbaatar et al., 2015; Vinyals et al., 2015; Graves et al., 2014; 2016) or by promoting behaviour that resembles message-passing algorithms as it is the case in Graph Neural Networks and

related architectures (Scarselli et al., 2009; Li et al., 2016; Battaglia et al., 2018). Another approach is to provide neural network building blocks that are specialized to solve some types of combinatorial problems such as satisfiability (SAT) instances (Wang et al., 2019), mixed integer programs (Ferber et al., 2019), sparse inference (Niculae et al., 2018), or submodular maximization (Tschiatschek et al., 2018). A related mindset of learning inputs to an optimization problem gave rise to the "predict-and-optimize" framework and its variants (Elmachtoub & Grigas, 2017; Demirovic et al., 2019; Mandi et al., 2019). Some works have directly addressed the question of learning combinatorial optimization algorithms such as the TRAVELING-SALESMAN-PROBLEM in (Bello et al., 2017) or its vehicle routing variants (Nazari et al., 2018). A recent approach also learns combinatorial algorithms via a clustering proxy (Wilder et al., 2019).

There are also efforts to bridge the gap in the opposite direction; to use deep learning methods to improve state-of-the-art combinatorial solvers, typically by learning (otherwise hand-crafted) heuristics. Some works have again targeted the TRAVELING-SALESMAN-PROBLEM (Kool et al., 2019; Deudon et al., 2018; Bello et al., 2017) as well as other NP-Hard problems (Li et al., 2018). Also, more general solvers received some attention; this includes SAT-solvers (Selsam & Bjørner, 2019; Selsam et al., 2019), integer programming solvers (often with learning branch-and-bound rules) (Khalil et al., 2016; Balcan et al., 2018; Gasse et al., 2019) and SMT-solvers (satisfiability modulo theories)(Balunovic et al., 2018).

## 3  METHOD

Let us first formalize the notion of a combinatorial solver. We expect the solver to receive continuous input $w \in W \subseteq \mathbb{R}^N$ (e.g. edge weights of a fixed graph) and return discrete output $y$ from some finite set $Y$ (e.g. all traveling salesman tours on a fixed graph) that minimizes some cost $\mathbf{c}(w, y)$ (e.g. length of the tour). More precisely, the solver maps

$$w \mapsto y(w) \quad \text{such that} \quad y(w) = \arg\min_{y \in Y} \mathbf{c}(w, y). \tag{1}$$

We will restrict ourselves to objective functions $\mathbf{c}(w, y)$ that are **linear** , namely $\mathbf{c}(w, y)$ may be represented as

$$\mathbf{c}(w, y) = w \cdot \phi(y) \quad \text{for } w \in W \text{ and } y \in Y \tag{2}$$

in which $\phi \colon Y \to \mathbb{R}^N$ is an injective representation of $y \in Y$ in $\mathbb{R}^N$. For brevity, we omit the mapping $\phi$ and instead treat elements of $Y$ as discrete points in $\mathbb{R}^N$.

Note that such definition of a solver is still very general as there are **no assumptions on the set of constraints or on the structure of the output space $Y$.**

**Example 1** (Encoding shortest-path problem). *If $G = (V, E)$ is a given graph with vertices $s, t \in V$, the combinatorial solver for the $(s, t)$-SHORTEST-PATH would take edge weights $w \in W = \mathbb{R}^{|E|}$ as input and produce the shortest path $y(w)$ represented as $\phi(y) \subseteq \{0, 1\}^{|E|}$ an indicator vector of the selected edges. The cost function is then indeed the inner product $\mathbf{c}(w, y) = w \cdot \phi(y)$.*

The task to solve during back-propagation is the following. We receive the gradient $\mathrm{d}L/\mathrm{d}y$ of the global loss $L$ with respect to solver output $y$ at a given point $\hat{y} = y(\hat{w})$. We are expected to return $\mathrm{d}L/\mathrm{d}w$, the gradient of the loss with respect to solver input $w$ at a point $\hat{w}$.

Since $Y$ is finite, there are only finitely many values of $y(w)$. In other words, this function of $w$ is **piecewise constant** and the gradient is identically zero or does not exist (at points of jumps). This should not come as a surprise; if one does a small perturbation to edge weights of a graph, one *usually* does not change the optimal TSP tour and *on rare occasions* alters it drastically. This has an important consequence:

> The fundamental problem with differentiating through combinatorial solvers is not the lack of differentiability; the gradient exists *almost everywhere*. However, this gradient is a constant zero and as such is unhelpful for optimization.

Accordingly, we will *not rely* on standard techniques for gradient estimation (see (Mohamed et al., 2019) for a comprehensive survey).

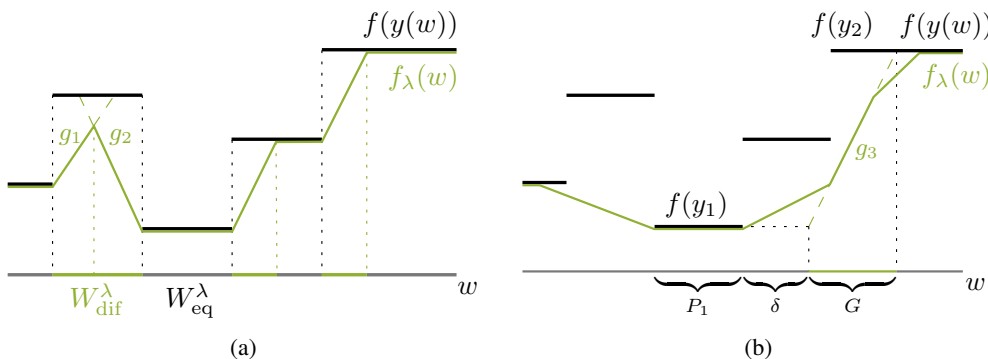

Figure 2: Continuous interpolation of a piecewise constant function. (a) $f_\lambda$ for a small value of $\lambda$; the set $W_{\text{eq}}^\lambda$ is still substantial and only two interpolators $g_1$ and $g_2$ are incomplete. Also, all interpolators are 0-interpolators. (b) $f_\lambda$ for a high value of $\lambda$; most interpolators are incomplete and we also encounter a $\delta$-interpolator $g_3$ (between $y_1$ and $y_2$) which attains the value $f(y_1)$ $\delta$-away from the set $P_1$. Despite losing some local structure for high $\lambda$, the gradient of $f_\lambda$ is still informative.

First, we simplify the situation by considering the linearization $f$ of $L$ at the point $\hat{y}$. Then for

$$f(y) = L(\hat{y}) + \frac{\mathrm{d}L}{\mathrm{d}y}(\hat{y}) \cdot (y - \hat{y}) \quad \text{we have} \quad \frac{\mathrm{d}f\big(y(w)\big)}{\mathrm{d}w} = \frac{\mathrm{d}L}{\mathrm{d}w}$$

and therefore it suffices to focus on differentiating the piecewise constant function $f\big(y(w)\big)$.

If the piecewise constant function at hand was arbitrary, we would be forced to use zero-order gradient estimation techniques such as computing finite differences. These require prohibitively many function evaluations particularly for high-dimensional problems.

However, the function $f\big(y(w)\big)$ is a result of a minimization process and it is known that for smooth spaces $Y$ there are techniques for such "differentiation through argmin" (Schmidt & Roth, 2014; Samuel & Tappen, 2009; Foo et al., 2008; Domke, 2012; Amos et al., 2017; Amos & Kolter, 2017). It turns out to be possible to build – with different mathematical tools – a viable discrete analogy. In particular, we can efficiently **construct a function** $f_\lambda(w)$, **a continuous interpolation** of $f\big(y(w)\big)$, whose gradient we return (see Fig. 2). The hyper-parameter $\lambda > 0$ controls the trade-off between "informativeness of the gradient" and "faithfulness to the original function".

Before diving into the formalization, we present the final algorithm as listed in Algo. 1. It is simple to implement and the backward pass indeed only runs the solver once on modified input. Providing the justification, however, is not straightforward, and it is the subject of the rest of the section.

---

**Algorithm 1** Forward and Backward Pass

---

**function** FORWARDPASS($\hat{w}$)
   $\hat{y} := \textbf{Solver}(\hat{w})$      // $\hat{y} = y(\hat{w})$
   **save** $\hat{w}$ and $\hat{y}$ for backward pass
   **return** $\hat{y}$

**function** BACKWARDPASS($\frac{\mathrm{d}L}{\mathrm{d}y}(\hat{y})$, $\lambda$)
   **load** $\hat{w}$ and $\hat{y}$ from forward pass
   $w' := \hat{w} + \lambda \cdot \frac{\mathrm{d}L}{\mathrm{d}y}(\hat{y})$
   *// Calculate perturbed weights*
   $y_\lambda := \textbf{Solver}(w')$
   **return** $\nabla_w f_\lambda(\hat{w}) := -\frac{1}{\lambda}\big[\hat{y} - y_\lambda\big]$
   *// Gradient of continuous interpolation*

---

### 3.1    CONSTRUCTION AND PROPERTIES OF $f_\lambda$

Before we give the exact definition of the function $f_\lambda$, we formulate several requirements on it. This will help us understand why $f_\lambda(w)$ is a reasonable replacement for $f\big(y(w)\big)$ and, most importantly, why its gradient captures changes in the values of $f$.

**Property A1.** For each $\lambda > 0$, $f_\lambda$ is continuous and piecewise affine.

The second property describes the trade-off induced by changing the value of $\lambda$. For $\lambda > 0$, we define sets $W_{\text{eq}}^\lambda$ and $W_{\text{dif}}^\lambda$ as the sets where $f(y(w))$ and $f_\lambda(w)$ coincide and where they differ, i.e.

$$W_{\text{eq}}^\lambda = \left\{ w \in W : f_\lambda(w) = f(y(w)) \right\} \quad \text{and} \quad W_{\text{dif}}^\lambda = W \setminus W_{\text{eq}}^\lambda.$$

**Property A2.** The sets $W_{\text{dif}}^\lambda$ are monotone in $\lambda$ and they vanish as $\lambda \to 0^+$, i.e.

$$W_{\text{dif}}^{\lambda_1} \subseteq W_{\text{dif}}^{\lambda_2} \quad \text{for } 0 < \lambda_1 \leq \lambda_2 \quad \text{and} \quad W_{\text{dif}}^\lambda \to \emptyset \quad \text{as } \lambda \to 0^+.$$

In other words, Property A2 tells us that $\lambda$ controls the size of the set where $f_\lambda$ deviates from $f$ and where $f_\lambda$ has meaningful gradient. This behaviour of $f_\lambda$ can be seen in Fig. 2.

In the third and final property, we want to capture the interpolation behavior of $f_\lambda$. For that purpose, we define a $\delta$-*interpolator* of $f$. We say that $g$, defined on a set $G \subset W$, is a $\delta$-interpolator of $f$ between $y_1$ and $y_2 \in Y$, if

- $g$ is non-constant affine function;
- the image $g(G)$ is an interval with endpoints $f(y_1)$ and $f(y_2)$;
- $g$ attains the boundary values $f(y_1)$ and $f(y_2)$ at most $\delta$-far away from where $f(y(w))$ does. In particular, there is a point $w_k \in G$ for which $g(w_k) = f(y_k)$ and $\text{dist}(w_k, P_k) \leq \delta$, where $P_k = \{w \in W : y(w) = y_k\}$, for $k = 1, 2$.

In the special case of a 0-interpolator $g$, the graph of $g$ connects (in a topological sense) two components of the graph of $f(y(w))$. In the general case, $\delta$ measures *displacement* of the interpolator (see also Fig. 2 for some examples). This displacement on the one hand loosens the connection to $f(y(w))$ but on the other hand allows for less local interpolation which might be desirable.

**Property A3.** The function $f_\lambda$ consists of finitely many (possibly incomplete) $\delta$-interpolators of $f$ on $W_{\text{dif}}^\lambda$ where $\delta \leq C\lambda$ for some fixed $C$. Equivalently, the *displacement* is linearly controlled by $\lambda$.

Intuitively, the consequence of Property A3 is that $f_\lambda$ has reasonable gradients everywhere since it consists of elementary affine interpolators.

For defining the function $f_\lambda$, we need a solution of a perturbed optimization problem

$$y_\lambda(w) = \arg\min_{y \in Y} \{\mathbf{c}(w, y) + \lambda f(y)\}. \tag{3}$$

**Theorem 1.** *Let $\lambda > 0$. The function $f_\lambda$ defined by*

$$f_\lambda(w) = f(y_\lambda(w)) - \frac{1}{\lambda}\left[\mathbf{c}(w, y(w)) - \mathbf{c}(w, y_\lambda(w))\right] \tag{4}$$

*satisfies Properties A1, A2, A3.*

Let us remark that already the continuity of $f_\lambda$ is not apparent from its definition as the first term $f(y_\lambda(w))$ is still a piecewise constant function. Proof of this result, along with geometrical description of $f_\lambda$, can be found in section A.2. Fig. 3 visualizes $f_\lambda$ for different values if $\lambda$.

Now, since $f_\lambda$ is ensured to be differentiable, we have

$$\nabla f_\lambda(w) = -\frac{1}{\lambda}\left[\frac{d\mathbf{c}}{dw}(w, y(w)) - \frac{d\mathbf{c}}{dw}(w, y_\lambda(w))\right] = -\frac{1}{\lambda}\left[y(w) - y_\lambda(w)\right]. \tag{5}$$

The second equality then holds due to (2). We then return $\nabla f_\lambda$ as a loss gradient.

**Remark 1.** *The roots of the method we propose lie in loss-augmented inference. In fact, the update rule from (5) (but not the function $f_\lambda$ or any of its properties) was already proposed in a different context in (Hazan et al., 2010; Song et al., 2016) and was later used in (Lorberbom et al., 2018; Mohapatra et al., 2018). The main difference to our work is that only the case of $\lambda \to 0^+$ is recommended and studied, which in our situation computes the correct but uninformative zero gradient. Our analysis implies that* **larger values of $\lambda$ are not only sound but even preferable***. This will be seen in experiments where we use values $\lambda \approx 10 - 20$.*

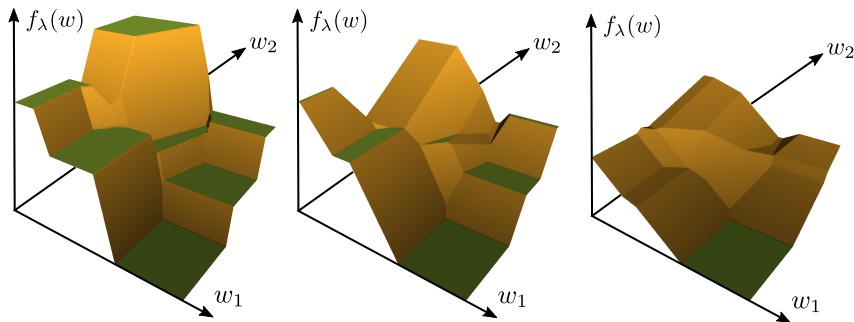

Figure 3: Example $f_\lambda$ for $w \in \mathbb{R}^2$ and $\lambda = 3, 10, 20$ (left to right). As $\lambda$ changes, the interpolation $f_\lambda$ is less faithful to the piecewise constant $f(y(w))$ but provides reasonable gradient on a larger set.

## 3.2 EFFICIENT COMPUTATION OF $f_\lambda$

Computing $y_\lambda$ in (3) is the only potentially expensive part of evaluating (5). However, the linear interplay of the cost function and the gradient trivially gives a resolution.

**Proposition 1.** *Let $\hat{w} \in W$ be fixed. If we set $w' = \hat{w} + \lambda \frac{dL}{dy}(\hat{y})$, we can compute $y_\lambda$ as*

$$y_\lambda(\hat{w}) = \underset{y \in Y}{\arg \min} \, \mathbf{c}(w', y).$$

In other words, $y_\lambda$ is the output of calling the solver on input $w'$.

## 4 EXPERIMENTS

In this section, we **experimentally validate a proof of concept**: that architectures containing exact blackbox solvers (with backward pass provided by Algo. 1) can be trained by standard methods.

Table 1: Experiments Overview.

| Graph Problem | Solver | Solver instance size | Input format |
|---|---|---|---|
| Shortest path | Dijkstra | up to 900 vertices | (image) up to $240 \times 240$ |
| Min Cost PM | Blossom V | up to 1104 edges | (image) up to $528 \times 528$ |
| Traveling Salesman | Gurobi | up to 780 edges | up to 40 images ($20 \times 40$) |

To that end, we solve three synthetic tasks as listed in Tab. 1. These tasks are designed to mimic practical examples from Section 2 and solving them anticipates a two-stage process: 1) extract suitable features from raw input, 2) solve a combinatorial problem over the features. The dimensionalities of input and of intermediate representations also aim to mirror practical problems and are chosen to be prohibitively large for zero-order gradient estimation methods. Guidelines of setting the hyperparameter $\lambda$ are given in section A.1.

We include the performance of ResNet18 (He et al., 2016) as a sanity check to demonstrate that the constructed datasets are too complex for standard architectures.

**Remark 2.** *The included solvers have very efficient implementations and do not severely impact runtime. All models train in under two hours on a single machine with 1 GPU and no more than 24 utilized CPU cores. Only for the large TSP problems the solver's runtime dominates.*

### 4.1 WARCRAFT SHORTEST PATH

**Problem input and output.** The training dataset for problem $SP(k)$ consists of 10000 examples of randomly generated images of terrain maps from the Warcraft II tileset (Guyomarch, 2017). The maps have an underlying grid of dimension $k \times k$ where each vertex represents a terrain with a fixed cost that is unknown to the network. The shortest (minimum cost) path between top left and bottom right vertices is encoded as an indicator matrix and serves as a label (see also Fig. 4). We consider datasets $SP(k)$ for $k \in \{12, 18, 24, 30\}$. More experimental details are provided in section A.3.

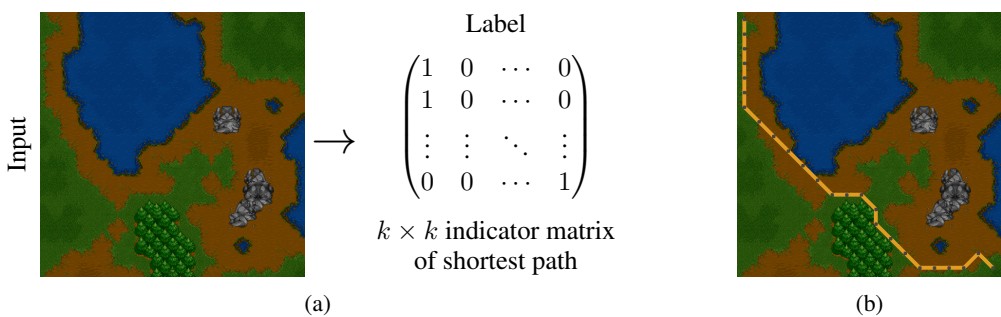

Label

$$\begin{pmatrix} 1 & 0 & \cdots & 0 \\ 1 & 0 & \cdots & 0 \\ \vdots & \vdots & \ddots & \vdots \\ 0 & 0 & \cdots & 1 \end{pmatrix}$$

$k \times k$ indicator matrix
of shortest path

(a)  (b)

Figure 4: The SP($k$) dataset. (a) Each input is a $k \times k$ grid of tiles corresponding to a Warcraft II terrain map, the respective label is a the matrix indicating the shortest path from top left to bottom right. (b) is a different map with correctly predicted shortest path.

**Architecture.** An image of the terrain map is presented to a convolutional neural network which outputs a $k \times k$ grid of vertex costs. These costs are then the input to the Dijkstra algorithm to compute the predicted shortest path for the respective map. The loss used for computing the gradient update is the Hamming distance between the true shortest path and the predicted shortest path.

**Results.** Our method learns to predict the shortest paths with high accuracy and generalization capability, whereas the ResNet18 baseline unsurprisingly fails to generalize already for small grid sizes of $k = 12$. Since the shortest paths in the maps are often nonunique (i.e. there are multiple shortest paths with the same cost), we report the percentage of shortest path predictions that have optimal cost. The results are summarized in Tab. 2.

Table 2: **Results for Warcraft shortest path.** Reported is the accuracy, i.e. percentage of paths with the optimal costs. Standard deviations are over five restarts.

| | Embedding Dijkstra | | ResNet18 | |
| --- | --- | --- | --- | --- |
| $k$ | Train % | Test % | Train % | Test % |
| 12 | $99.7 \pm 0.0$ | $96.0 \pm 0.3$ | $100.0 \pm 0.0$ | $23.0 \pm 0.3$ |
| 18 | $98.9 \pm 0.2$ | $94.4 \pm 0.2$ | $99.9 \pm 0.0$ | $0.7 \pm 0.3$ |
| 24 | $97.8 \pm 0.2$ | $94.4 \pm 0.6$ | $100.0 \pm 0.0$ | $0.0 \pm 0.0$ |
| 30 | $97.4 \pm 0.1$ | $94.0 \pm 0.3$ | $95.6 \pm 0.5$ | $0.0 \pm 0.0$ |

## 4.2 GLOBE TRAVELING SALESMAN PROBLEM

**Problem input and output.** The training dataset for problem TSP($k$) consists of 10000 examples where the input for each example is a $k$-element subset of fixed 100 country flags and the label is the shortest traveling salesman tour through the capitals of the corresponding countries. The optimal tour is represented by its adjacency matrix (see also Fig. 5). We consider datasets TSP($k$) for $k \in \{5, 10, 20, 40\}$.

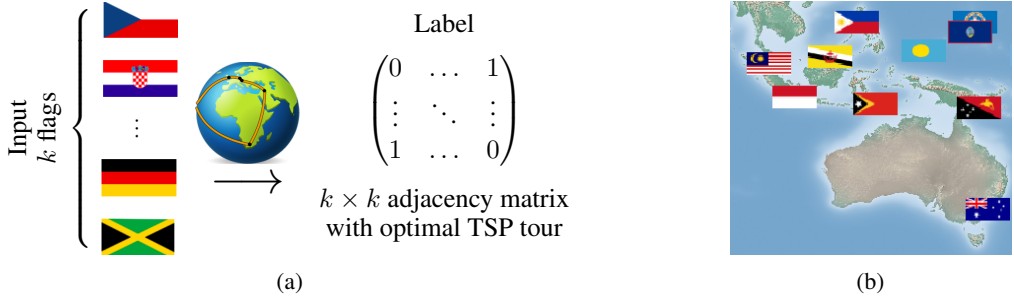

Label

$$\begin{pmatrix} 0 & \cdots & 1 \\ \vdots & \ddots & \vdots \\ 1 & \cdots & 0 \end{pmatrix}$$

$k \times k$ adjacency matrix
with optimal TSP tour

(a)  (b)

Figure 5: The TSP($k$) problem. (a) illustrates the dataset. Each input is a sequence of $k$ flags and the corresponding label is the adjacency matrix of the optimal TSP tour around the corresponding capitals. (b) displays the learned locations of 10 country capitals in southeast Asia and Australia, accurately recovering their true position.

**Architecture.** Each of the $k$ flags is presented to a convolutional network that produces $k$ three-dimensional vectors. These vectors are projected onto the unit sphere in $\mathbb{R}^3$; a representation of the globe. The TSP solver receives a matrix of pairwise distances of the $k$ computed locations. The loss of the network is the Hamming distance between the true and the predicted TSP adjacency matrix. The architecture is expected to learn the correct representations of the flags (i.e. locations of the respective countries' capitals on Earth, up to rotations of the sphere). The employed Gurobi solver optimizes a mixed-integer programming formulation of TSP using the cutting plane method (Marchand et al., 2002) for lazy sub-tour elimination.

**Results.** This architecture not only learns to extract the correct TSP tours but also learns the correct representations. Quantitative evidence is presented in Tab. 3, where we see that the learned locations generalize well and lead to correct TSP tours also on the test set and also on somewhat large instances (note that there are $39! \approx 10^{46}$ admissible TSP tours for $k = 40$). The baseline architecture only memorizes the training set. Additionally, we can extract the suggested locations of world capitals and compare them with reality. To that end, we present Fig. 5b, where the learned locations of 10 capitals in Southeast Asia are displayed.

Table 3: **Results for Globe TSP.** Reported is the full tour accuracy. Standard deviations are over five restarts.

| | Embedding TSP Solver | | ResNet18 | |
|---|---|---|---|---|
| $k$ | Train % | Test % | Train % | Test % |
| 5 | $99.8 \pm 0.0$ | $99.2 \pm 0.1$ | $100.0 \pm 0.0$ | $1.9 \pm 0.6$ |
| 10 | $99.8 \pm 0.1$ | $98.7 \pm 0.2$ | $99.0 \pm 0.1$ | $0.0 \pm 0.0$ |
| 20 | $99.1 \pm 0.1$ | $98.4 \pm 0.4$ | $98.8 \pm 0.3$ | $0.0 \pm 0.0$ |
| 40 | $97.4 \pm 0.2$ | $96.7 \pm 0.4$ | $96.9 \pm 0.3$ | $0.0 \pm 0.0$ |

### 4.3 MNIST MIN-COST PERFECT MATCHING

**Problem input and output.** The training dataset for problem $\mathrm{PM}(k)$ consists of 10000 examples where the input to each example is a set of $k^2$ digits drawn from the MNIST dataset arranged in a $k \times k$ grid. For computing the label, we consider the underlying $k \times k$ grid graph (without diagonal edges) and solve a MIN-COST-PERFECT-MATCHING problem, where edge weights are given simply by reading the two vertex digits as a two-digit number (we read downwards for vertical edges and from left to right for horizontal edges). The optimal perfect matching (i.e. the label) is encoded by an indicator vector for the subset of the selected edges, see example in Fig. 6.

**Architecture.** The grid image is the input of a convolutional neural network which outputs a grid of vertex weights. These weights are transformed into edge weights as described above and given to the solver. The loss function is Hamming distance between solver output and the true label.

**Results.** The architecture containing the solver is capable of good generalizations suggesting that the correct representation is learned. The performance is good even on larger instances and despite the presence of noise in supervision – often there are many optimal matchings. In contrast, the ResNet18 baseline only achieves reasonable performance for the simplest case $\mathrm{PM}(4)$. The results are summarized in Tab. 4.

Table 4: **Results for MNIST Min-cost perfect matching.** Reported is the accuracy of predicting an optimal matching. Standard deviations are over five restarts.

| | Embedding Blossom V | | ResNet18 | |
|---|---|---|---|---|
| $k$ | Train % | Test % | Train % | Test % |
| 4 | $99.97 \pm 0.01$ | $98.32 \pm 0.24$ | $100.0 \pm 0.0$ | $92.5 \pm 0.3$ |
| 8 | $99.95 \pm 0.04$ | $99.92 \pm 0.01$ | $100.0 \pm 0.0$ | $8.3 \pm 0.8$ |
| 16 | $99.02 \pm 0.84$ | $99.06 \pm 0.57$ | $100.0 \pm 0.0$ | $0.0 \pm 0.0$ |
| 24 | $95.63 \pm 5.49$ | $92.06 \pm 7.97$ | $96.1 \pm 0.5$ | $0.0 \pm 0.0$ |

## 5 DISCUSSION

We provide a unified mathematically sound algorithm to embed combinatorial algorithms into neural networks. Its practical implementation is straightforward and training succeeds with standard deep learning techniques. The two main branches of future work are: 1) exploring the potential of newly enabled architectures, 2) addressing standing real-world problems. The latter case requires

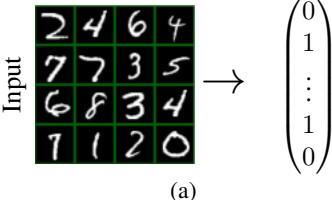 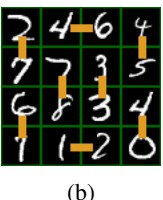

(a)                                                                    (b)

Figure 6: Visualization of the PM dataset. (a) shows the case of PM(4). Each input is a $4 \times 4$ grid of MNIST digits and the corresponding label is the indicator vector for the edges in the min-cost perfect matching. (b) shows the correct min-cost perfect matching output from the network. The cost of the matching is 348 ($46 + 12$ horizontally and $27 + 45 + 40 + 67 + 78 + 33$ vertically).

embedding approximate solvers (that are common in practice). This breaks some of our theoretical guarantees but given their strong empirical performance, the fusion might still work well in practice.

## ACKNOWLEDGEMENT

We thank the International Max Planck Research School for Intelligent Systems (IMPRS-IS) for supporting Marin Vlastelica. We acknowledge the support from the German Federal Ministry of Education and Research (BMBF) through the Tbingen AI Center (FKZ: 01IS18039B). Additionally, we would like to thank Paul Swoboda and Alexander Kolesnikov for valuable feedback on an early version of the manuscript.

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

## A  APPENDIX

### A.1  GUIDELINES FOR SETTING THE VALUES OF $\lambda$.

In practice, $\lambda$ has to be chosen appropriately, but we found its exact choice uncritical (no precise tuning was required). Nevertheless, note that $\lambda$ should cause a noticeable disruption in the optimization problem from equation (3), otherwise it is too likely that $y(w) = y_\lambda(w)$ resulting in a zero gradient. In other words, $\lambda$ should roughly be of the magnitude that brings the two terms in the definition of $w'$ in Prop. 1 to the same order:

$$\lambda \approx \frac{\langle w \rangle}{\left\langle \frac{\mathrm{d}L}{\mathrm{d}y} \right\rangle}$$

where $\langle \cdot \rangle$ stands for the average. This again justifies that $\lambda$ is a **true hyperparameter** and that there is no reason to expect values around $\lambda \to 0^+$.

### A.2  PROOFS

**Proof of Proposition 1.** Let us write $L = L(\hat{y})$ and $\nabla L = \frac{\mathrm{d}L}{\mathrm{d}y}(\hat{y})$, for brevity. Thanks to the linearity of $\mathbf{c}$ and the definition of $f$, we have

$$\mathbf{c}(\hat{w}, y) + \lambda f(y) = \hat{w}y + \lambda\big(L + \nabla L(y - \hat{y})\big) = (\hat{w} + \lambda\nabla L)y + \lambda L - \lambda\nabla L\hat{y} = \mathbf{c}(w', y) + \mathbf{c}_0,$$

where $\mathbf{c}_0 = \lambda L - \lambda\nabla L\hat{y}$ and $w' = \hat{w} + \lambda\nabla L$ as desired. The conclusion about the points of minima then follows. □

Before we prove Theorem 1, we make some preliminary observations. To start with, due to the definition of the solver, we have the fundamental inequality

$$\mathbf{c}(w, y) \geq \mathbf{c}\big(w, y(w)\big) \quad \text{for every } w \in W \text{ and } y \in Y. \tag{6}$$

**Observation 1.** *The function* $w \mapsto \mathbf{c}\big(w, y(w)\big)$ *is continuous and piecewise linear.*

**Proof.** Since $\mathbf{c}$'s are linear and distinct, $\mathbf{c}\big(w, y(w)\big)$, as their pointwise minimum, has the desired properties. □

Analogous fundamental inequality

$$\mathbf{c}(w, y) + \lambda f(y) \geq \mathbf{c}(w, y_\lambda(w)) + \lambda f(y_\lambda(w)) \quad \text{for every } w \in W \text{ and } y \in Y \qquad (7)$$

follows from the definition of the solution to the optimization problem (3).

A counterpart of Observation 1 reads as follows.

**Observation 2.** *The function* $w \mapsto \mathbf{c}(w, y_\lambda(w)) + \lambda f(y_\lambda(w))$ *is continuous and piecewise affine.*

**Proof.** The function under inspection is a pointwise minimum of distinct affine functions $w \mapsto \mathbf{c}(w, y) + \lambda f(y)$ as $y$ ranges $Y$. $\qquad \square$

As a consequence of above-mentioned fundamental inequalities, we obtain the following two-sided estimates on $f_\lambda$.

**Observation 3.** *The following inequalities hold for* $w \in W$

$$f(y_\lambda(w)) \leq f_\lambda(w) \leq f(y(w)).$$

**Proof.** Inequality (6) implies that $\mathbf{c}(w, y(w)) - \mathbf{c}(w, y_\lambda(w)) \leq 0$ and the first inequality then follows simply from the definition of $f_\lambda$. As for the second one, it suffices to apply (7) to $y = y(w)$. $\qquad \square$

Now, let us introduce few notions that will be useful later in the proofs. For a fixed $\lambda$, $W$ partitions into maximal connected sets $P$ on which $y_\lambda(w)$ is constant (see Fig. 7). We denote this collection of sets by $\mathcal{W}_\lambda$ and set $\mathcal{W} = \mathcal{W}_0$.

For $\lambda \in \mathbb{R}$ and $y_1 \neq y_2 \in Y$, we denote

$$F_\lambda(y_1, y_2) = \{w \in W : c(w, y_1) + \lambda f(y_1) = c(w, y_2) + \lambda f(y_2)\}.$$

We write $F(y_1, y_2) = F_0(y_1, y_2)$, for brevity. For technical reasons, we also allow negative values of $\lambda$ here.

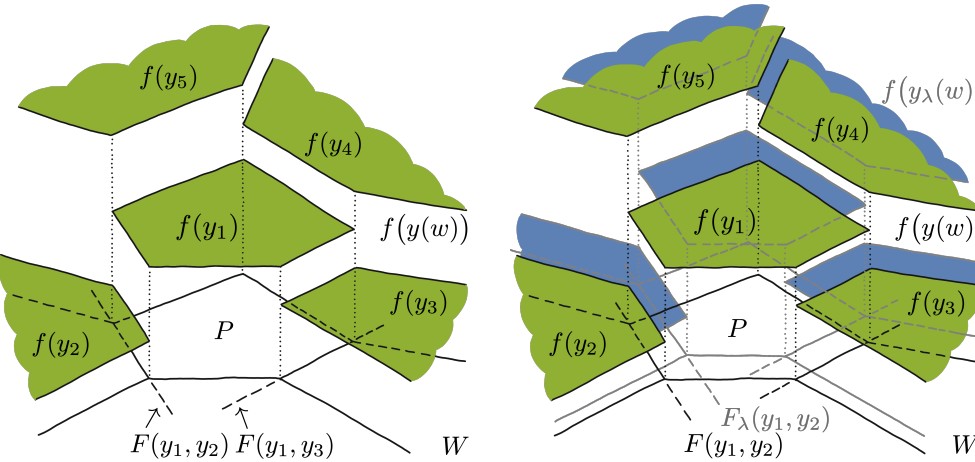

(a) The situation for $\lambda = 0$. We can see the polytope $P$ on which $y(w)$ attains $y_1 \in Y$. The boundary of $P$ is composed of segments of lines $F(y_1, y_k)$ for $k = 2, \ldots, 5$.

(b) The same situation is captured for some relatively small $\lambda > 0$. Each line $F_\lambda(y_1, y_k)$ is parallel to its corresponding $F(y_1, y_k)$ and encompasses a convex polytope in $\mathcal{W}_\lambda$.

Figure 7: The family $\mathcal{W}_\lambda$ of all maximal connected sets $P$ on which $y_\lambda$ is constant.

Note, that if $W = \mathbb{R}^N$, then $F_\lambda$ is a hyperplane since $\mathbf{c}$'s are linear. In general, $W$ may just be a proper subset of $\mathbb{R}^N$ and, in that case, $F_\lambda$ is just the restriction of a hyperplane onto $W$. Consequently, it may happen that $F_\lambda(y_1, y_2)$ will be empty for some pair of $y_1, y_2$ and some $\lambda \in \mathbb{R}$. To emphasize this fact, we say "hyperplane in $W$". Analogous considerations should be taken into account for all other linear objects. The note "in $W$" stands for the intersection of these linear object with the set $W$.

**Observation 4.** *Let $P \in \mathcal{W}_\lambda$ and let $y_\lambda(w) = y$ for $w \in P$. Then $P$ is a convex polytope in $W$, where the facets consist of parts of finitely many hyperplanes $F_\lambda(y, y_k)$ in $W$ for some $\{y_k\} \subset Y$.*

**Proof.** Assume that $W = \mathbb{R}^N$. The values of $y_\lambda$ may only change on hyperplanes of the form $F_\lambda(y, y')$ for some $y' \in Y$. Then $P$ is an intersection of corresponding half-spaces and therefore $P$ is a convex polytope. If $W$ is a proper subset of $\mathbb{R}^N$ the claim follows by intersecting all the objects with $W$. $\square$

**Observation 5.** *Let $y_1, y_2 \in Y$ be distinct. If nonempty, the hyperplanes $F(y_1, y_2)$ and $F_\lambda(y_1, y_2)$ are parallel and their distance is equal to $|\lambda| K(y_1, y_2)$, where*

$$K(y_1, y_2) = \frac{|f(y_1) - f(y_2)|}{\|y_1 - y_2\|}.$$

**Proof.** If we define a function $c(w) = \mathbf{c}(w, y_1) - \mathbf{c}(w, y_2) = w(y_1 - y_2)$ and a constant $C = f(y_2) - f(y_1)$, then our objects rewrite to

$$F(y_1, y_2) = \{w \in W : c(w) = 0\} \quad \text{and} \quad F_\lambda(y_1, y_2) = \{w \in W : c(w) = \lambda C\}.$$

Since $c$ is linear, these sets are parallel and $F(y_1, y_2)$ intersects the origin. Thus, the required distance is the distance of the hyperplane $F_\lambda(y_1, y_2)$ from the origin, which equals to $|\lambda C|/\|y_1 - y_2\|$. $\square$

As the set $Y$ is finite, there is a uniform upper bound $K$ on all values of $K(y_1, y_2)$. Namely

$$K = \max_{\substack{y_1, y_2 \in Y \\ y_1 \neq y_2}} K(y_1, y_2). \tag{8}$$

### A.2.1 PROOF OF THEOREM 1

**Proof of Property A1.** Now, Property A1 follows, since

$$f_\lambda(w) = \frac{1}{\lambda} \left[ \mathbf{c}(w, y_\lambda(w)) + \lambda f(y_\lambda(w)) \right] - \frac{1}{\lambda} \mathbf{c}(w, y(w))$$

and $f_\lambda$ is a difference of continuous and piecewise affine functions. $\square$

**Proof of Property A2.** Let $0 < \lambda_1 \leq \lambda_2$ be given. We show that $W_{\text{eq}}^{\lambda_2} \subseteq W_{\text{eq}}^{\lambda_1}$ which is the same as showing $W_{\text{dif}}^{\lambda_1} \subseteq W_{\text{dif}}^{\lambda_2}$. Assume that $w \in W_{\text{eq}}^{\lambda_2}$, that is, by the definition of $W_{\text{eq}}^{\lambda_2}$ and $f_\lambda$,

$$\mathbf{c}(w, y(w)) + \lambda_2 f(y(w)) = \mathbf{c}(w, y_2) + \lambda_2 f(y_2), \tag{9}$$

in which we denoted $y_2 = y_{\lambda_2}(w)$. Our goal is to show that

$$\mathbf{c}(w, y(w)) + \lambda_1 f(y(w)) = \mathbf{c}(w, y_1) + \lambda_1 f(y_1), \tag{10}$$

where $y_1 = y_{\lambda_1}(w)$ as this equality then guarantees that $w \in W_{\text{eq}}^{\lambda_1}$. Observe that (7) applied to $\lambda = \lambda_1$ and $y = y(w)$, yields the inequality "$\geq$" in (10).

Let us show the reversed inequality. By Observation 3 applied to $\lambda = \lambda_1$, we have

$$f(y(w)) \geq f(y_1). \tag{11}$$

We now use (7) with $\lambda = \lambda_2$ and $y = y_1$, followed by equality (9) to obtain

$$\begin{aligned}
\mathbf{c}(w, y_1) + \lambda_1 f(y_1) &= \mathbf{c}(w, y_1) + \lambda_2 f(y_1) + (\lambda_1 - \lambda_2) f(y_1) \\
&\geq \mathbf{c}(w, y_2) + \lambda_2 f(y_2) + (\lambda_1 - \lambda_2) f(y_1) \\
&= \mathbf{c}(w, y(w)) + \lambda_2 f(y(w)) + (\lambda_1 - \lambda_2) f(y_1) \\
&= \mathbf{c}(w, y(w)) + \lambda_1 f(y(w)) + (\lambda_2 - \lambda_1) [f(y(w)) - f(y_1)] \\
&\geq \mathbf{c}(w, y(w)) + \lambda_1 f(y(w))
\end{aligned}$$

where the last inequality holds due to (11).

Next, we have to show that $W_{\text{dif}}^\lambda \to \emptyset$ as $\lambda \to 0^+$, i.e. that for almost every $w \in W$, there is a $\lambda > 0$ such that $w \notin W_{\text{dif}}^\lambda$. To this end, let $w \in W$ be given. We can assume that $y(w)$ is a unique solution of solver (1), since two solutions, say $y_1$ and $y_2$, coincide only on the hyperplane $F(y_1, y_2)$ in $W$, which is of measure zero. Thus, since $Y$ is finite, the constant

$$c = \min_{\substack{y \in Y \\ y \neq y(w)}} \left\{ \mathbf{c}(w, y) - \mathbf{c}(w, y(w)) \right\}$$

is positive. Denote

$$d = \max_{y \in Y} \left\{ f(y(w)) - f(y) \right\}. \tag{12}$$

If $d > 0$, set $\lambda < c/d$. Then, for every $y \in Y$ such that $f(y(w)) > f(y)$, we have

$$\lambda < \frac{\mathbf{c}(w, y) - \mathbf{c}(w, y(w))}{f(y(w)) - f(y)}$$

which rewrites

$$\mathbf{c}(w, y(w)) + \lambda f(y(w)) < \mathbf{c}(w, y) + \lambda f(y). \tag{13}$$

For the remaining $y$'s, (13) holds trivially for every $\lambda > 0$. Therefore, $y(w)$ is a solution of the minimization problem (3), whence $y_\lambda(w) = y(w)$. This shows that $w \in W_{\text{eq}}^\lambda$ as we wished. If $d = 0$, then $f(y(w)) \leq f(y)$ for every $y \in Y$ and (13) follows again. $\qquad\square$

**Proof of Property A3.** Let $y_1 \neq y_2 \in Y$ be given. We show that on the component of the set

$$\{w \in W : y(w) = y_1 \text{ and } y_\lambda(w) = y_2\} \tag{14}$$

the function $f_\lambda$ agrees with a $\delta$-interpolator, where $\delta \leq C\lambda$ and $C > 0$ is an absolute constant. The claim follows as there are only finitely many sets and their components of the form (14) in $W_{\text{dif}}^\lambda$.

Let us set

$$h(w) = \mathbf{c}(w, y_1) - \mathbf{c}(w, y_2) \quad \text{for } w \in W$$

and

$$g(w) = f(y_2) - \frac{1}{\lambda} h(w).$$

The condition on $\mathbf{c}$ tells us that $h$ is a non-constant affine function. It follows by the definition of $F(y_1, y_2)$ and $F_\lambda(y_1, y_2)$ that

$$h(w) = 0 \quad \text{if and only if} \quad w \in F(y_1, y_2) \tag{15}$$

and

$$h(w) = \lambda(f(y_2) - f(y_1)) \quad \text{if and only if} \quad w \in F_\lambda(y_1, y_2). \tag{16}$$

By Observation 5, the sets $F$ and $F_\lambda$ are parallel hyperplanes. Denote by $G$ the nonempty intersection of their corresponding half-spaces in $W$. We show that $g$ is a $\delta$-interpolator of $f$ on $G$ between $y_1$ and $y_2$, with $\delta$ being linearly controlled by $\lambda$.

We have already observed that $g$ is the affine function ranging from $f(y_1)$ – on the set $F_\lambda(y_1, y_2)$ – to $f(y_2)$ – on the set $F(y_1, y_2)$. It remains to show that $g$ attains both the values $f(y_1)$ and $f(y_2)$ at most $\delta$-far from the sets $P_1$ and $P_2$, respectively, where $P_k \in \mathcal{W}$ denotes a component of the set $\{w \in W : y(w) = y_k\}$, $k = 1, 2$.

Consider $y_1$ first. By Observation 4, there are $z_1, \ldots, z_\ell \in Y$, such that facets of $P_1$ are parts of hyperplanes $F(y_1, z_1), \ldots, F(y_1, z_\ell)$ in $W$. Each of them separates $W$ into two half-spaces, say $W_k^+$ and $W_k^-$, where $W_k^-$ is the half-space which contains $P_1$ and $W_k^+$ is the other one. Let us denote

$$c_k(w) = \mathbf{c}(w, y_1) - \mathbf{c}(w, z_k) \quad \text{for } w \in W \text{ and } k = 1, \ldots, \ell.$$

Every $c_k$ is a non-zero linear function which is negative on $W_k^-$ and positive on $W_k^+$. By the definition of $y_1$, we have

$$\mathbf{c}(w, y_1) + \lambda f(y_1) \leq \mathbf{c}(w, z_k) + \lambda f(z_k) \quad \text{for } w \in P_1 \text{ and for } k = 1, \ldots, \ell,$$

that is

$$c_k(w) \leq \lambda(f(z_k) - f(y_1)) \quad \text{for } w \in P_1 \text{ and for } k = 1, \ldots, \ell.$$

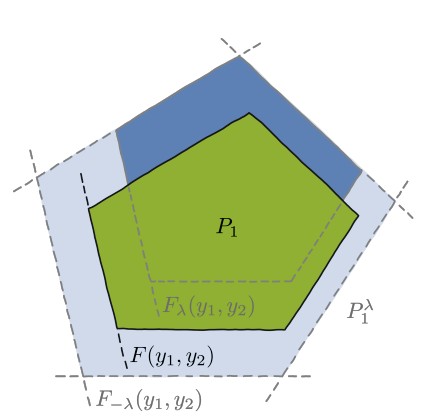
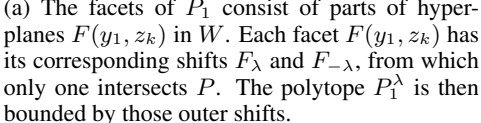
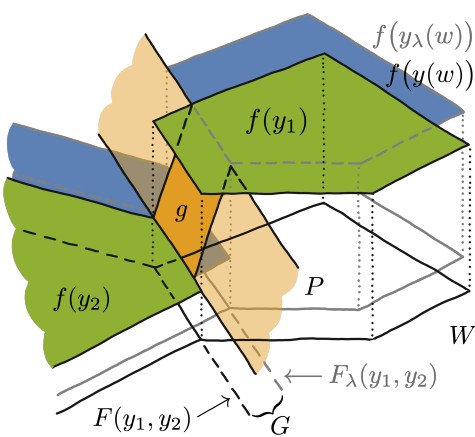

(a) The facets of $P_1$ consist of parts of hyperplanes $F(y_1, z_k)$ in $W$. Each facet $F(y_1, z_k)$ has its corresponding shifts $F_\lambda$ and $F_{-\lambda}$, from which only one intersects $P$. The polytope $P_1^\lambda$ is then bounded by those outer shifts.

(b) The interpolator $g$ attains the value $f(y_1)$ on a part of $F_\lambda(y_1, y_2)$ – a border of the domain $G$. The value $f(y_2)$ is attained on a part of $F(y_1, y_2)$ – the second border of the strip $G$.

Figure 8: The polytopes $P_1$ and $P_1^\lambda$ and the interpolator $g$.

Now, denote

$$W_k^\lambda = \left\{ w \in W : c_k(w) \leq \lambda \big| f(z_k) - f(y_1) \big| \right\} \quad \text{for } k = 1, \ldots, \ell.$$

Each $W_k^\lambda$ is a half-space in $W$ containing $W_k^-$ and hence $P_1$. Let us set $P_1^\lambda = \bigcap_{k=1}^\ell W_k^\lambda$. Clearly, $P_1 \subseteq P_1^\lambda$ (see Fig. 8). By Observation 5, the distance of the hyperplane $\{ w \in W : c_k(w) = \lambda \big| f(z_k) - f(y_1) \big| \}$ from $P_1$ is at most $\lambda K$, where $K$ is given by (8). Therefore, since all the facets of $P_1^\lambda$ are at most $\lambda K$ far from $P_1$, there is a constant $C$ such that each point of $P_1^\lambda$ is at most $C\lambda$ far from $P_1$.

Finally, choose any $w_1 \in P_1^\lambda \cap F_\lambda(y_1, y_2)$. By (16), we have $g(w_1) = f(y_1)$, and by the definition of $P_1^\lambda$, $w_1$ is no farther than $C\lambda$ away from $P_1$.

Now, let us treat $y_2$ and define the set $P_2^\lambda$ analogous to $P_1^\lambda$, where each occurrence of $y_1$ is replaced by $y_2$. Any $w_2 \in P_2^\lambda \cap F(y_1, y_2)$ has desired properties. Indeed, (15) ensures that $g(w_2) = f(y_2)$ and $w_2$ is at most $C\lambda$ far away from $P_2$. $\qquad\square$

## A.3 DETAILS OF EXPERIMENTS

### A.3.1 WARCRAFT SHORTEST PATH

The maps for the dataset have been generated with a custom random generation process by using 142 tiles from the Warcraft II tileset (Guyomarch, 2017). The costs for the different terrain types range from 0.8–9.2. Some example maps of size $18 \times 18$ are presented in Fig. 9a together with a histogram of the shortest path lengths. We used the first five layers of ResNet18 followed by a max-pooling operation to extract the latent costs for the vertices.

Optimization was carried out via Adam optimizer (Kingma & Ba, 2014) with scheduled learning rate drops dividing the learning rate by 10 at epochs 30 and 40. Hyperparameters and model details are listed in Tab. 5

Table 5: Experimental setup for Warcraft Shortest Path.

| k | Optimizer(LR) | Architecture | Epochs | Batch Size | $\lambda$ |
|---|---|---|---|---|---|
| 12, 18, 24, 30 | Adam($5 \times 10^{-4}$) | subset of ResNet18 | 50 | 70 | 20 |

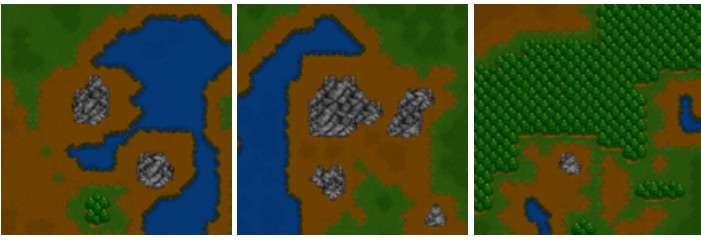 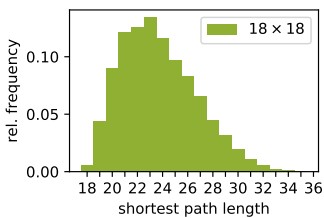

(a) Three random example maps.

(b) the shortest path distribution in the training set. All possible path lengths (18-35) occur.

Figure 9: Warcraft SP(18) dataset.

### A.3.2 MNIST MIN-COST PERFECT MATCHING

The dataset consists of randomly generated grids of MNIST digits that are sampled from a subset of 1000 digits of the full MNIST dataset. We trained a fully convolutional neural network with two convolutional layers followed by a max-pooling operation that outputs a $k \times k$ grid of vertex costs for each example. The vertex costs are transformed into the edge costs via the known cost function and the edge costs are then the inputs to the Blossom V solver (Edmonds, 1965) as implemented in (Kolmogorov, 2009).

Regarding the optimization procedure, we employed the Adam optimizer along with scheduled learning rate drops dividing the learning rate by 10 at epochs 10 and 20, respectively. Other training details are in Tab. 6. Lower batch sizes were used to reduce GPU memory requirements.

Table 6: Experimental setup for MNIST Min-cost Perfect Matching.

| k | Optimizer(LR) | Architecture [channels, kernel size, stride] | Epochs | Batch Size | $\lambda$ |
|---|---|---|---|---|---|
| 4, 8 | Adam($10^{-3}$) | $[[20, 5, 1], [20, 5, 1]]$ | 30 | 70 | 10 |
| 16 | Adam($10^{-3}$) | $[[50, 5, 1], [50, 5, 1]]$ | 30 | 40 | 10 |
| 24 | Adam($10^{-3}$) | $[[50, 5, 1], [50, 5, 1]]$ | 30 | 30 | 10 |

### A.3.3 GLOBE TRAVELING SALESMAN PROBLEM

For the Globe Traveling Salesman Problem we used a convolutional neural network architecture of three convolutional layers and two fully connected layers. The last layer outputs a vector of dimension $3k$ containing the $k$ 3-dimensional representations of the respective countries' capital cities. These representations are projected onto the unit sphere and the matrix of pairwise distances is fed to the TSP solver.

The high combinatorial complexity of TSP has negative effects on the loss landscape and results in many local minima and high sensitivity to random restarts. For reducing sensitivity to restarts, we set Adam parameters to $\beta_1 = 0.5$ (as it is done for example in GAN training (Radford et al., 2015)) and $\epsilon = 10^{-3}$.

The local minima correspond to solving planar TSP as opposed to spherical TSP. For example, if all cities are positioned to almost identical locations, the network can still make progress but it will never have the incentive to spread the cities apart in order to reach the global minimum. To mitigate that, we introduce a repellent force between epochs 15 and 30. In particular, we set

$$L_{\text{rep}} = \mathop{\mathbb{E}}_{i \neq j} e^{-\|x_i - x_j\|}$$

where $x_i \in \mathbb{R}^3$ for $i = 1, \ldots, k$ are the positions of the $k$ cities on the unit sphere. The regularization constants $C_k$ were chosen as $2.0, 3.0, 6.0$, and $20.0$ for $k \in \{5, 10, 20, 40\}$.

For fine-tuning we also introduce scheduled learning rate drops where we divide the learning rate by 10 at epochs 80 and 90.

Table 7: Experimental setup for the Globe Traveling Salesman Problem.

| k | Optimizer(LR) | Architecture [channels, kernel size, stride], linear layer size | Epochs | Batch Size | $\lambda$ |
|---|---|---|---|---|---|
| 5, 10, 20 | Adam($10^{-4}$) | $[[20, 4, 2], [50, 4, 2], 500]$ | 100 | 50 | 20 |
| 40 | Adam($5 \times 10^{-5}$) | $[[20, 4, 2], [50, 4, 2], 500]$ | 100 | 50 | 20 |

In Fig. 5b, we compare the true city locations with the ones learned by the hybrid architecture. Due to symmetries of the sphere, the architecture can embed the cities in any rotated or flipped fashion. We resolve this by computing "the most favorable" isometric transformation of the suggested locations. In particular, we solve the orthogonal Procrustes problem (Gower & Dijksterhuis, 2004)

$$R^* = \underset{R:R^T R=I}{\arg\min} \|RX - Y\|^2$$

where $X$ are the suggested locations, $Y$ the true locations, and $R^*$ the optimal transformation to apply. We report the resulting offsets in kilometers in Tab. 8.

Table 8: Average errors of city placement on the Earth.

| k | 5 | 10 | 20 | 40 |
|---|---|---|---|---|
| Location offset (km) | $69 \pm 11$ | $19 \pm 5$ | $11 \pm 5$ | $58 \pm 7$ |

### A.4 TRAVELING SALESMAN WITH AN APPROXIMATE SOLVER

Since approximate solvers often appear in practice where the combinatorial instances are too large to be solved exactly in reasonable time, we test our method also in this setup. In particular, we use the approximate solver (OR-Tools (ort, 2019)) for the Globe TSP. We draw two conclusions from the numbers presented below in Tab. 9.

 (i) The choice of the solver matters. Even if OR-Tools is fed with the ground truth representations (i.e. true locations) it does not achieve perfect results on the test set (see the right column). We expect, that also in practical applications, running a suboptimal solver (e.g. a differentiable relaxation) substantially reduces the maximum attainable performance.

 (ii) The suboptimality of the solver didn't harm the feature extraction – the point of our method. Indeed, the learned locations yield performance that is close to the upper limit of what the solver allows (compare the middle and the right column).

Table 9: **Perfect path accuracy** for Globe TSP using the approximate solver OR-Tools (ort, 2019). The maximal achievable performance is in the right column, where the solver uses the ground truth city locations.

| k | Embedding OR-tools Train % | Embedding OR-tools Test % | OR-tools on GT locations Test % |
|---|---|---|---|
| 5 | $99.8 \pm 0.0$ | $99.3 \pm 0.1$ | 100.0 |
| 10 | $84.3 \pm 0.2$ | $84.4 \pm 0.2$ | 88.6 |
| 20 | $49.2 \pm 0.2$ | $48.6 \pm 0.8$ | 54.4 |
| 40 | $14.6 \pm 0.1$ | $15.1 \pm 0.3$ | 15.2 |

