# OpenReview forum: "Differentiation of Blackbox Combinatorial Solvers"
_ICLR.cc/2020/Conference — Accept (Spotlight)_

### Official Review · AnonReviewer3 · 2019-10-22
**Official Blind Review #3**

**Rating:** 8

**Review:**

=== Summary ===
The authors propose a method for efficiently backpropagating through unmodified blackbox implementations of exact combinatorial solvers with linear objective functions.
The gradient of such exact combinatorial solvers exists almost everywhere but is zero. The authors remark that the loss has the same gradient wrt to the solver's input as its linearization around the solver's input.  They therefore propose to interpolate the loss' linearization with a continuous (piecewise affine) function and use the gradient of this interpolation to backpropagate through the solver. This gradient is obtained efficiently by simply calling the solver on a single perturbed input (the perturbation depends on the incoming gradient, ie the gradient of the loss wrt to the solver's output).
The authors further study the properties of this piecewise affine interpolation and characterize its interpolation behavior as a function of a hyperparameter which controls the trade-off between "how informative the gradient is" and "how faithful the interpolation is to the original solver".
The authors validate their method with experiments on synthetic tasks that have both a visual processing aspect and a combinatorial aspect:
    - Shortest Path on Warcraft II terrain maps
    - TSP between country capitals where the inputs to the convnet are country flags
    - Min-cost perfect matching from Mnist digits.
Specifically, they feed the output of a convnet to the relevant solver (depending on the task) and learn end-to-end by backpropagating through the solver with their proposed method. They show that their method successfully solves the tasks where baseline ConvNet architectures fail.

=== Recommendation ===

This paper addresses an important problem and presents a novel approach.

Methods for combining combinatorial optimization algorithms and machine learning usually rely on modifiying or relaxing the combinatorial problem itself which prevents using solvers as-is.
In contrast, the presented method allows to efficiently backpropagate through unmodified implementations of blackbox exact solvers with a linear objective. AFAIK this is the first method that allows this.

A weakness of the paper is that the experiments only validate proof of concept (as noted by the authors). They are small-scale and only compare against conventional ConvNets baselines (as opposed to other approaches to backpropagate through relaxed combinatorial problems).
Additionally, the characterization of the interpolation (whose gradient is used) doesn't directly explain why the gradient of the interpolation is a reasonable choice.

Overall, I recommend for acceptance.

=== Questions / Comments ===
- The authors show properties related to the interpolation behavior of the proposed interpolation function. What is the actual point/benefit of satisfying these properties? Are there arguments for why this is important besides the experimental results?  Is the point that since lambda controls how "faithful vs informative" the gradient is , there must be a range of values for lambda for which the method works?
- It would be interesting to have experiments with non-exact solvers
- It would be interesting to optimize directly for the combinatorial objective in the experiments (using a policy gradient for example) rather than perform supervised learning on the solutions.
- Consider adding related work subsection on argmin optimization and meta-learning.






**Experience Assessment:**

I have published one or two papers in this area.

**Review Assessment: Checking Correctness Of Derivations And Theory:**

I assessed the sensibility of the derivations and theory.

**Review Assessment: Checking Correctness Of Experiments:**

I assessed the sensibility of the experiments.

**Review Assessment: Thoroughness In Paper Reading:**

I read the paper thoroughly.

---

> ### Author Response · Authors · 2019-11-11
> **Response to Reviewer #3**
>
> Thank you for your comments.
>
> Regarding approximate solvers and baselines, please refer to the joint part of our response.
>
> == Meaning of Theoretical Guarantees ==
>
> Theorem 1 indeed does not give any usual type of a guarantee. To our knowledge, there are however no established techniques for evaluating gradients suggested for piecewise constant functions (any kind of comparison to the true zero gradient misses the point).
> In this uncharted territory, our intention was to give a theoretical description and guarantees about the *process* sketched in Figure 3.
>
> The technical insights embedded in the proofs might also allow for proving different type of guarantees. What would be a convincing statement about piecewise constant function interpolation that Reviewer 3 would like to see?
>
> We can certainly make some improvements on the presentation side. For example, connect Property A2 to Figure 3 (green regions shrink with lambda) or offer a more intuitive interpretation of Property A3 (it suggests that gradients of f\_lambda are reasonable everywhere -- as elementary interpolators have certainly reasonable gradients).
>
> == Type of Supervision ==
>
> We agree that supervision based on the value of the combinatorial objective is natural for example in reinforcement learning scenarios and we will look into it in the future. The full supervision we use is however not artificial. The motivation comes from computer vision tasks where the ground truth assignment is typically known (e.g. segmentation, stereo matching, pose estimation).
>
> == Additional Related Work ==
>
> Driven by maintaining focus on the main message, we primarily included literature at the intersection of deep learning and combinatorial optimization in the related work section. The literature that is relevant from the optimization point of view is cited throughout the method section (differentiation through argmin is also discussed). If the reviewer is missing references to concrete works, we will include them. We agree that if the list of optimization references grows by more than a couple of papers, it would be reasonable to introduce a new subsection of Section 2, and we will do so in that case.

---

### Official Review · AnonReviewer1 · 2019-10-23
**Official Blind Review #1**

**Rating:** 8

**Review:**

This paper proposes a straightforward method for training black box solvers of a restricted kind (namely those with inputs in R^n and linear cost functions). The proposed algorithm is tested on path finding, the travelling salesman problem, and a min-cost-perfect-matching problem, with promising results.

I would recommend accepting this paper. It is a well written paper with a novel idea supported by good experimental results.

The caveat is that I did not have the time to thoroughly review all the mathematical details. From a high-level they looked correct, and the math is sufficiently illustrated with figures and examples that it is easy for a reader to follow in detail given enough time.

The main shortcomings I see are that there are no experimental results comparing this method against any existing results; the authors do compare against their own ResNet18 implementation, but this is not ideal.

I found the discussion a bit cryptic: Why are approximate solvers needed for real-world problems? Are there no real-world problems where exact solvers are still applicable?

**Experience Assessment:**

I do not know much about this area.

**Review Assessment: Checking Correctness Of Derivations And Theory:**

I assessed the sensibility of the derivations and theory.

**Review Assessment: Checking Correctness Of Experiments:**

I assessed the sensibility of the experiments.

**Review Assessment: Thoroughness In Paper Reading:**

I read the paper at least twice and used my best judgement in assessing the paper.

---

> ### Author Response · Authors · 2019-11-11
> **Response to Reviewer #1**
>
> Thank you for your comments and positive appraisal of our paper.
>
> == Discussion on Approximate Solvers ==
>
> The remark about approximate solvers is particularly aimed at computer vision applications (graph matching, multicut etc.). The combinatorial instances in such applications are large and exact solvers become impractical (e.g. Gurobi solver spends significant computation time proving optimality of an already known solution). We will clarify this in the final version of this work.
>
> See also our common response for baselines and an additional experiment with an approximate solver.

---

### Official Review · AnonReviewer2 · 2019-10-24
**Official Blind Review #2**

**Rating:** 8

**Review:**

This paper shows how end-to-end learning can be done through
combinatorial solvers by using the derivative of
continuous surrogate function in the backward pass.
One elegant part of the method is that no modification
or relaxation is done to the combinatorial solver in
the forward pass and that the backward pass just requires
another call to the blackbox solver.

The idea of constructing continuous surrogate functions
and using them for differentiating through solvers with
piecewise-constant output spaces is thought-provoking and
I can see it inspiring many new directions of work.
For example looking at Figure 2 for intuition, one could
imagine other ways of making the solution space continuous.
The solution space of linear programs over continuous spaces,
as considered in [Elmachtoub & Grigas], the Sudoku example in
[Amos & Kolter], and related papers, is also piecewise constant and
it seems like a similar method could be used to bring more
informative derivative information to linear programs ---
have you considered this as a future direction?

One of my concerns with this work is that the ResNet baseline in the
experimental results seems like too much of a straw man for the tasks.
I do not see why they should have the capacity to generalize well.
This paper shows the ResNet baseline achieve near-zero
test accuracy but doesn't compare to other relevant baselines
that are mentioned in the related work section:
for example [Bello et al, Deudon et al., Kool et al.] for the TSP.

And one smaller comment: If one wanted to squeeze the performance even
more, would starting the training process with a large \lamdba
and annealing it to zero help?

----

Elmachtoub, A. N., & Grigas, P. Smart "predict, then optimize". arXiv 2017.

**Experience Assessment:**

I have published in this field for several years.

**Review Assessment: Checking Correctness Of Derivations And Theory:**

I assessed the sensibility of the derivations and theory.

**Review Assessment: Checking Correctness Of Experiments:**

I assessed the sensibility of the experiments.

**Review Assessment: Thoroughness In Paper Reading:**

I read the paper at least twice and used my best judgement in assessing the paper.

---

> ### Author Response · Authors · 2019-11-11
> **Response to Reviewer #2**
>
> Thank you for your assessment. Please also refer to our common answer above.
>
> == Linear Programs  ==
>
> The distinction to make about linear programs is the output format. If the linear program seeks to find an integral solution (ILPs) our method is applicable out of the box. We agree this opens interesting directions for future work.  However, if continuous solutions are required, the function at hand is no longer piecewise constant and other methods (such as [Amos \& Kolter]) may be preferable in terms of gradient estimation/computation. Also note that our method, in its current form, is designed to optimize an objective and not decide feasibility, as is the case for Sudoku. It takes a deeper thought whether we can generalize the method in that direction.
>
> Also, thank you for pointing out [Elmachtoub, A. N., \& Grigas, P. Smart]; we were not aware of this work and will include a reference for the final version.
>
> == Lambda Adaptation ==
>
> We deliberately did not include any kind of lambda scheduling into the paper as we wanted to keep the method in its purest possible form. However, the proposed (and other) ideas are very inviting and we are currently looking into them in ongoing work. Surprisingly, it seems that only marginal improvements are possible and the constant lambda baseline is quite strong.

---

> > ### Comment · AnonReviewer2 · 2019-11-15
> > **Reviewer Response**
> >
> > Thanks for the additional details and clarifications! It is surprising the constant \lambda baseline is quite strong. I have gone through the other reviews and discussions and this thread and agree with R3 that this is a clear accept and have also updated my score to an 8.

---

### Author Response · Authors · 2019-11-11
**Common Answer**

We thank the reviewers for their comments.

== Choice of Baselines ==

Our main aim was to enable solving various types of combinatorial problems from raw inputs without making any concessions on the combinatorial side. From that perspective, there is no clear baseline to compare with (other than maybe zero order methods). We believe, this puts us in a similar position as works [Wang and Kolter] and [Amos and Kolter] who provided similar building blocks for convex optimization and satisfiability. We actually built on their experimental design -- compare against a ResNet on clean synthetic tasks -- however, with larger dimensionalities in both the raw images and the solver inputs. The main purpose of the ResNet baseline is to make sure the datasets do not contain easily exploitable features.

Given that there is a volume of work at the intersection of deep learning and combinatorial optimization -- as we also list in Section 2 -- it seems hard to believe that there is no appropriate baseline. Let us briefly explain why for example the works [Bello et al, Deudon et al., Kool et al.] are not comparable to ours. We found similar mismatches with the other cited literature.

The works [Bello et al, Deudon et al., Kool et al.] aim to compete with the dedicated solvers; purely on the combinatorial side (see their experimental sections). They are not aspiring to be neural network building blocks. In fact, they are not even fully differentiable as the underlying reinforcement learning algorithm executes a sequence of discrete actions (i.e. the same piecewise constant structure emerges) to find the TSP tour. We do not see any natural way for differentiating these pipelines other than casting them as blackbox solvers and using our method.

Having said all of this, we do not claim (and never have) that our experimental section is a decisive proof of performance; this remains to be seen on more complex real-world applications. At this point, we claim a proof of concept with a broad potential of follow-up applications.

== Additional Experiment ==

We would like to propose to include the results of running our method with an approximate solver. For this we use the Google OR-Tools solver in the TSP experiment.
We draw two conclusions from the numbers presented below.

1) The choice of the solver matters. Even if OR-Tools is fed with the ground truth representations (i.e. true locations) it does not achieve perfect results on the test set (see the right column). We expect, that also in practical applications, running a suboptimal solver (e.g. a differentiable relaxation) substantially reduces the maximum attainable performance.

2) The suboptimality of the solver didn't harm the feature extraction -- the point of our method. Indeed, the learned locations yield performance that is close to the upper limit of what the solver allows (compare the middle and the right column).

                             Accuracy of perfect paths
     Embedding OR-Tools Solver   OR-Tools on GT representation
 k            Train%    Test%                                 Test %
 5            99.8        99.3                                    100.0
 10          84.3        84.4                                    88.6
 20          49.2        48.6                                    54.4
 40          14.6        15.1                                    15.2

Would you recommend to include this in the paper?

---

### Comment · AnonReviewer3 · 2019-11-11
**Addressing the authors' replies**

After reading the author's replies, I have changed my score to 8 as I believe this paper is a clear accept.

I still think the paper is a bit weak experimentally (for example, the authors could have applied the method presented in Bello et al with a small convnet to extract (x,y,z) from the country flags) but the authors have presented their work in a fair and honest light and addressed potential concerns.

- Meaning of theoretical guarantees: I am also not aware of techniques to evaluate such gradients. I believe the presentation could be a bit improved: the flow of the paper (during my first pass) made me expect some sort of theoretical guarantee.
- My comments on the type of supervision and related work are suggested as potential improvements of the paper.
- I recommend adding results with non-exact solvers as they show that the feature extraction process also works with non-exact solvers

---

### Decision · Program_Chairs · 2019-12-19

**Decision:**

Accept (Spotlight)

**Comment:**

This paper proposes a method for efficiently training neural networks combined with blackbox implementations of exact combinatorial solvers.

Reviewers and AC agree that it is a well written paper with a novel idea supported by good experimental results. Experimental results are of small scale and can be further improved, but the authors acknowledged this aspect well.

Hence, I recommend acceptance.